# Immunohistochemical Expression Patterns of Inflammatory Cells Involved in Chronic Hyperplastic Candidosis

**DOI:** 10.3390/pathogens8040232

**Published:** 2019-11-12

**Authors:** Ailish Williams, David Williams, Helen Rogers, Xiaoqing Wei, Michael Lewis, Sue Wozniak, Damian Farnell, Adam Jones

**Affiliations:** 1School of Dentistry, Cardiff University, Heath Park, Cardiff CF14 4XY, UK; Williamsaj24@cardiff.ac.uk (A.W.); weix1@cardiff.ac.uk (X.W.); lewismao@cardiff.ac.uk (M.L.); Farnelld@cardiff.ac.uk (D.F.); jonesa108@cardiff.ac.uk (A.J.); 2Bristol Dental School, Lower Maudlin Street, Bristol BS1 3NU, UK; Helen.rogers2@UHBristol.nhs.uk; 3Dental Hospital, University Hospital of Wales, Heath Park, Cardiff CF14 4XY, UK; sue.wozniak@wales.nhs.uk

**Keywords:** inflammatory cells, chronic hyperplastic candidosis, immunohistochemistry, T lymphocytes, B lymphocytes

## Abstract

The profile of the inflammatory cell infiltrate in chronic hyperplastic candidosis (CHC) was determined in oral mucosal biopsies by immunohistochemistry. One tonsillar tissue section was included as an immunohistochemistry control, whilst squamous papilloma (*n* = 4) with secondary *Candida* infection was used as *Candida* controls. Oral lichen planus tissues (*n* = 10) provided negative controls for *Candida* presence, as well as positive controls for inflammation. Immunohistochemistry employed antibodies specific for CD3^+^ (T lymphocytes), CD4^+^ (T helper cells), CD8^+^ (cytotoxic T cells), and CD20^+^ (B lymphocytes). Manual counting of stained cells from digitised images determined the proportion of each cell type relative to the total number of cells, and these were assessed in the mucosa, the epithelium, and the lamina propria. The mean proportion of CD3^+^ cells was significantly higher than CD20^+^ cells in all tissue types. For CHC, the mean proportion of CD3^+^ cells in entire tissues was 15.6%, with the highest proportion in the lamina propria (32.6%) compared with the epithelium (3.9%). CD20^+^ cells were in much lower proportions (1.8%) in CHC, with the highest proportion (3.6%) in the lamina propria. T lymphocytes were predominately CD4^+^ cells (9.0%) compared with CD8^+^ cells (4.4%). CD4^+^ cells were most prevalent in the lamina propria (23.1%) compared with the epithelium (mean = 3.2%). From these results, it was concluded that the immune response invoked by *Candida* in CHC is primarily driven by the T helper cells.

## 1. Introduction

*Candida albicans* is a commensal fungus of humans, where it typically resides on the skin and mucosal surfaces without detriment to health. However, in debilitated individuals, including those with immune deficiency, *C. albicans* can cause a range of opportunistic infections, collectively referred to as candidoses. Whilst these infections are mostly superficial, primarily effecting the oral and vaginal mucosa, in severely immunocompromised patients, serious systemic infections can arise, which have mortality rates approaching 50% [1]. 

Several clinical presentations of oral candidoses are recognised, including acute and chronic pseudomembranous candidosis, acute erythematous candidosis, chronic erythematous candidosis, and chronic hyperplastic candidosis (CHC) [2]. These infections typically arise following changes in the oral environment, often associated with weakened or immature immune systems [3]. An appropriately functioning immune response is therefore essential in protecting the host against candidosis [4].

The focus of this research was CHC, which is associated with premalignant changes (oral epithelial dysplasia), although it is not clear whether premalignancy occurs because of the *Candida* infection itself [5]. CHC presents as white plaque lesions on the commissures of the oral mucosa or buccal mucosa, and the lateral border of the tongue [6]. Unlike other oral candidoses, where only surface colonisation by the fungus occurs, in CHC, invasion of *Candida* into the keratinised layer of the oral epithelium is evident. Histology of CHC lesions is essential for diagnosis, and along with *Candida* invasion, typically reveals an inflammatory cell infiltrate with hyperplasia of the oral epithelium [6,7]. There is debate over interaction between immune cells when coordinating host responses to *Candida,* and this is further complicated by differential responses to yeast and hyphal forms of *C. albicans*. When *Candida* invades the oral mucosa, a sequence of cellular interactions occurs, ultimately leading to phagocytosis and removal of the invading *Candida* [8]. It is thought that Toll-Like Receptors (TLRs) and C-lectin receptors on dendritic cells (DCs) recognise pathogen-associated molecular patterns (PAMPs) on the surface of *Candida* [8,9]. The DCs then phagocytose the *Candida* and migrate to draining lymph nodes, where the antigen is presented to naïve T cells. T cells bind to the DCs via the presented antigen, and major histocompatibility complex (MHC) [10]. The result of interaction between the DC with the naïve T cell is release of specific cytokines leading to clonal expansion of the T cell and their trafficking to the infection site. T cells are thought to be essential in the orchestration of cell immunity [4]. In addition, recent studies have evidenced a role for oral epithelial cells themselves to detect and, indeed, discriminate between yeast and hyphal forms of *C. albicans* through pattern recognition receptors. This may subsequently instigate both the induction of pro-inflammatory and antifungal responses [11,12]. 

All T cells express CD3 (cluster of differentiation 3), which is a cell surface protein cluster that acts as a receptor for activation. Within the CD3^+^ T cell population, some T cells are CD4^+^ and these are commonly referred to as T helper cells. Other CD3^+^ T cells are CD8^+^ and are often referred to as cytotoxic T cells. CD8^+^ T cells can be resident within the epithelium of normal mucosa, and occasionally CD4^+^ T cells are found in the corium, which in the oral mucosa is most frequently termed the lamina propria. T helper (CD4^+^) cells produce cytokines that ‘help’ the immune response [9,13], whilst the primary function of CD8^+^ cells (cytotoxic T cells) is to kill viral-infected cells [13]. Specific types of CD4^+^ T helper (Th) cell responses may occur and are mediated by subsets of CD4^+^ T cells termed Th1, Th2, Th17, and T regulatory cells (Tregs) [10]. Th-1 responses are pro-inflammatory [12] and are modulated by the anti-inflammatory effects of Th2 responses. Th-17 responses are associated with IL-17A, IL-17F, IL-21, and IL-22 cytokines and are strongly linked with protection against fungal disease [12,14]. In addition to T cells, B-lymphocytes (CD20^+^) can be present in the lamina propria and these mature into plasma cells, which could play a role in defence against *Candida* through production of antibodies. The most important mucosal defence against *Candida* infection is undoubtedly cell-mediated immunity (involving T cells) and innate immunity (involving macrophages, neutrophils, and cytotoxic T cells) [15]. There remains controversy about the role of antibodies in protecting against mucosal candidosis, although protection against systemic candidosis has been demonstrated [16]. 

To date, the immunohistochemical analyses of inflammatory cells in CHC has not been widely studied. Previous research examined expression of CD3^+^ and CD20^+^ cells in CHC, but a lack of commercially available antibodies which functioned on fixed tissue at the time of that study meant analyses of the CD4^+^ and CD8^+^ T cell subsets were not possible [7]. This present study includes this analysis, thus further enhancing our knowledge of host immune responses to *Candida* in CHC.

## 2. Results

### 2.1. Immunohistochemical Analysis

Analyses were undertaken by two assessors, and intraclass correlation coefficient testing revealed consistent scoring (kappa score 0.990). Figure 1 illustrates the typical immunohistochemical staining of the targeted cell types for the different tissue types. Tonsil tissue was included as an immunohistochemistry control, whilst squamous papilloma with secondary *Candida* infection served as *Candida* controls. The oral lichen planus tissue provided a negative control for *Candida* presence, as well as a positive control for inflammation.

### 2.2. Proportion of CD3^+^ and CD20^+^ Cells in Tissues 

The mean percentages (compared with total cells) of CD3^+^ and CD20^+^ cells in tissues are presented in Table 1 and Figure 2, and Table 2 and Figure 3, respectively. For all tissues, there was a significantly (*p* < 0.0001) higher proportion of T cells compared with B cells. CD3^+^ cells detected in all tissues, although a significant difference (*p* = 0.0281) in the proportion of these cells occurred between tissues types. The highest proportion of CD3^+^ cells was in oral lichen planus (33.3%) and lowest in squamous papilloma (4.9%). In CHC, the highest proportion of CD3^+^ cells was in the lamina propria (32.6%), with only a small proportion occurring in the epithelium (3.9%). 

All tissues had low levels of CD20^+^ cells (Table 2 and Figure 3), and there was no significant difference (*p* = 0.3908) between tissue types. The highest proportion of CD20^+^ cells occurred in oral lichen planus (2.9%) and lowest in squamous papilloma (0.6%). 

For CHC, a mean proportion of CD3^+^ cells of 15.6% was evident, compared with 1.7% for CD20^+^ cells. In oral lichen planus, a significantly (*p* < 0.0001) higher proportion of CD3^+^ cells (33.3%) than CD20^+^ cells (2.9%) was evident.

### 2.3. Proportion of CD4^+^ and CD8^+^ Cells in Tissues 

Table 3 and Figure 4, and Table 4 and Figure 5, present the mean proportions of CD4^+^ and CD8^+^ cells in tissues, respectively. There were significant differences in proportions of CD4^+^ cells (*p* = 0.0430) for the different tissues. CD4^+^ cells had highest prevalence in the lamina propria of oral lichen planus (54.6%) compared with CHC (23.0%). The lowest proportion of CD4^+^ cells was in squamous papilloma (7.4%). Significant differences in proportions of CD4^+^ cells between tissue types occurred in the epithelium (*p* = 0.008), where these cells were more prevalent in CHC (3.2%) compared with both oral lichen planus (0.7%) and squamous papilloma (1.0%). 

The proportion of CD8^+^ cells was also significantly different between tissue types (*p* < 0.001). CD8^+^ cells were most prevalent in the lamina propria of oral lichen planus (54.4%) compared with CHC (15.0%) and squamous papilloma (1.1%). No significant differences in the number of CD8^+^ cells in the epithelium was evident. In the case of CHC, a significantly (*p* < 0.0001) higher proportion of CD4^+^ cells (9.0%) was evident compared with CD8^+^ cells (4.4%).

## 3. Discussion

The primary aim of this study was to characterise the inflammatory cell infiltrate in CHC, which, to date, has not been widely investigated. CHC is an important form of oral candidosis, given its potential association with malignant transformation at lesional sites. This study applied immunohistochemistry to sections of archival CHC biopsy tissue and targeted four types of immune cells, namely CD3^+^ (T cells), CD20^+^ (B cells), CD4^+^ (T helper cells), and CD8^+^ (cytotoxic T cells) cells. Using this approach, it was possible, for the first time, to identify subsets of T cells in CHC. 

As anticipated, given its chronic inflammatory nature [17], oral lichen planus had the highest levels of both T and B cells. The results did, however, show that the proportion of CD4^+^ cells was higher (34.0%) than CD8^+^ cells (29.2%), which could reflect level of disease activity. It is not uncommon for squamous papilloma to be secondarily infected by *Candida* and, interestingly, limited host immune responses tend to occur in such cases [18]. This finding was supported by the results of the present study, where squamous papilloma tissues had the lowest levels of both T and B cells. In contrast, CHC tissues had higher proportions of both T helper (CD4^+^) and cytotoxic T cells (CD8^+^) in the lamina propria compared with squamous papilloma, suggesting that the host immune response differed between the conditions and was likely dependent on *Candida* infection.

The importance of CD4^+^ cells in protection against oral candidosis is highlighted by higher disease incidence in individuals with reduced CD4^+^ cell numbers [19]. CHC does, however, occur in people who are not thought to have lowered CD4^+^ cell numbers or function. Chromophore analysis of tissue from oropharyngeal candidosis has previously indicated that the majority of T cells were CD8^+^, rather than CD4^+^ cells [19]. In contrast, in immunodeficient transgenic mice with oral and gastrointestinal candidosis, the immune response was primarily mediated by CD4^+^ Th1 cells [20]. 

A previous study of CHC inflammatory cells used polyclonal rabbit antibodies, and analysed infiltration of T cells, B cells, and plasma cells in CHC [7]. The findings of this present study were similar, with higher CD3^+^ cell numbers relative to CD20^+^ cells being encountered. There was also agreement between the two studies regarding the higher density of the inflammatory cell infiltrate in the lamina propria compared with the epithelium. However, this present study was also able to determine the type of T cell present in the inflammatory infiltrate and found it to be predominantly driven by T helper cells rather than cytotoxic T cells. Ascertaining the subtypes of these T helper cells was not undertaken, but future study could focus on cytokine and interleukin (IL) expression by the T helper (Th) cells, which would be indicative Th1 (interferon-γ), Th2 (IL-4 and IL-5), Th17 (IL-17 and IL-22), and Treg (Foxp3 and IL-35) cells. As plasma cells containing IgG are also known to be in abundance in the lamina propria in oral candidosis [7], detection of CD79a and CD138 positive cells could offer improved understanding of the plasma cell infiltrate in these infections.

## 4. Materials and Methods 

### 4.1. Tissue Samples

Prior to commencement of this research, tissue biopsies from patients attending the Cardiff University School of Dentistry were obtained for diagnostic purposes. Pathological diagnoses for the tissues was established by the resident consultant pathologist. Ethical approval for this research was obtained (IRAS approved for Project ID: 136258) and permission also provided by the Cardiff and Vale University Health Board R&D (approval received 14/DEN/5953).

An alphanumeric code was used to blind researchers to patient ID and diagnostic tissue type during the study. After data were analysed, the tissue conditions were revealed so the data could be interpreted with respect to their pathology. The tissue types studied (Table 5) were chronic hyperplastic candidosis (*n* = 20), squamous papilloma (*n* = 4, positive control for *Candida* with secondary *Candida* colonisation), oral lichen planus (*n* = 10, negative control for *Candida* and positive control for inflammation, a non-*Candida*-associated mucocutaneous inflammatory condition), and tonsil (*n* = 1, positive control for inflammation and appropriate antibody reaction). 

### 4.2. Immunohistochemistry

Tissue sections were initially stained with haematoxylin and eosin (HE) using an automated slide staining system (Linistain, Thermo Scientific, Paisley, UK). Following HE staining, tissues were stained using the primary antibodies against human CD3, CD20, CD4, and CD8. Antibodies were all mouse monoclonal antibodies, with the exception of CD3 antibody, which was a rabbit antibody. All antibodies were obtained from Dako and were used without dilution to label tissues for 20 min. Antibody detection used the Dako Flex Envision™ ^+^ kit as recommended by the manufacturer. Negative controls included all tissue processing steps, apart from inclusion of primary antibody. Images of stained sections were obtained and digitised using an Aperio system (Leica Microsystems Ltd, Milton Keynes, UK). 

### 4.3. Histopathological Analysis of Stained Tissues

Using the Aperio Imagescope, histopathological analysis of the stained tissue sections was performed. For each tissue section, digital ‘demarcation’ was made between the epithelium and the lamina propria/corium (Figure 6). The tissue section was then further divided into equal portions across the epithelium and lamina propria. In each portion, estimation was made of the percentage of cells stained (relative to the total cell population) with CD3^+^, CD4^+^, CD8^+^, or CD20^+^ specific antibodies. Analysis was performed for all tissue sections by two researchers.

### 4.4. Statistical Analysis

The median average over repeated measurements within each slide was found for the epithelial, lamina propria, and ‘total’ (i.e., both epithelium and lamina propria) tissues separately in order to form the data for the main analysis (i.e., for each slide). Data were then explored initially by using descriptive statistics (mean, median, and standard deviations) evaluated over all slides and by using graphical box plots with dot plots of the "raw" data superimposed. Normality was assessed by using histograms, normal plots, and the Shapiro–Wilk and Kolgorov–Smirnov tests in SPSS V25. As non-normality could not be ruled out in many cases, differences between groups were assessed by using non-parametric one-way ANOVA (Kruskal–Wallis test), followed by Dunn’s post-hoc test in all cases by using the statistical software package R V3.30.

## 5. Conclusions

Understanding the host immune response in CHC is important, as CHC is potentially a pre-malignant condition, and this information could serve as both a diagnostic tool for CHC and also allow us to monitor the host for potential immune factors associated with CHC. From the research, significant differences in levels of CD3^+^, CD4^+^, CD8^+^, and CD20^+^ cells were found between the three tissue types studied. The highest levels of inflammatory cells occurred in oral lichen planus, and the lowest in squamous papilloma. CHC was shown to predominantly be a T helper cell response in the lamina propria, as opposed to a cytotoxic T cell response. Further research is needed to ascertain the subsets of these T helper cells.

## Figures and Tables

**Figure 1 pathogens-08-00232-f001:**
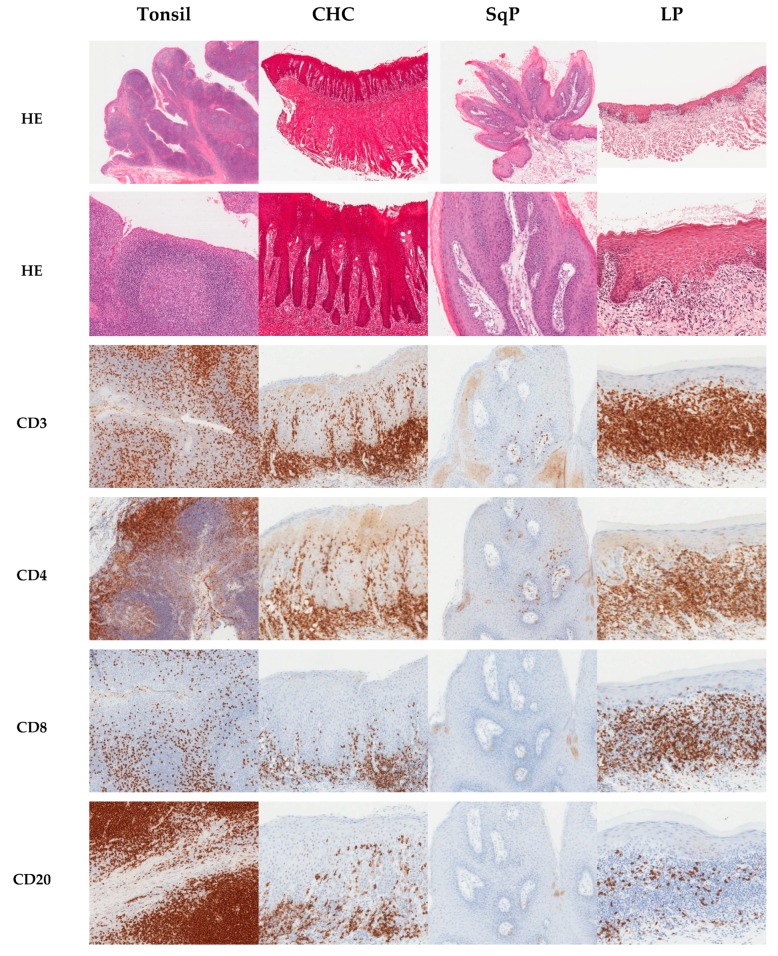
Composite image of haematoxylin and eosin (HE) and immunohistochemical staining of sections of tonsil, chronic hyperplastic candidosis (CHC), squamous papilloma (SqP), and oral lichen planus (LP). Immunohistochemical staining was for T cells (CD3^+^), T helper cells (CD4^+^), cytotoxic T cells (CD8^+^), and B cells (CD20^+^).^.^ Original magnification x100, except for first row of HE stained sections, which was x20.

**Figure 2 pathogens-08-00232-f002:**
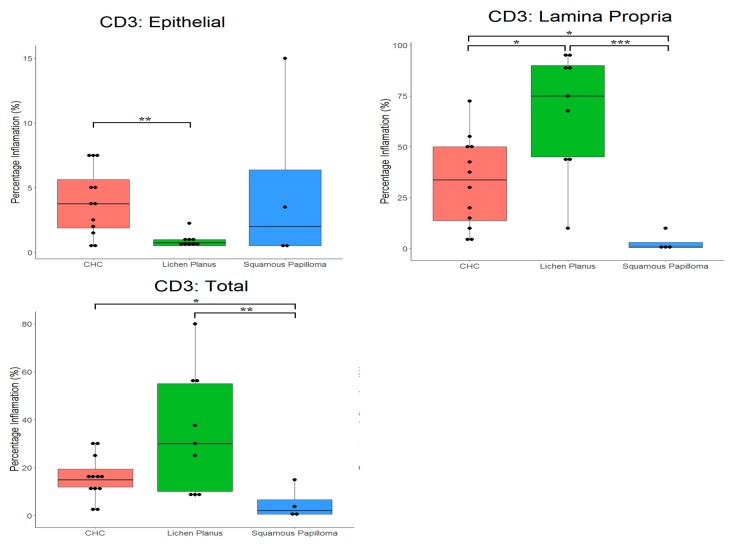
Box plots with the ‘raw’ data points superimposed, showing percentage of T cells (CD3^+^) in tissue types. CHC, chronic hyperplastic candidosis. * indicates *p* < 0.05, ** indicates *p* < 0.01, *** indicates *p* < 0.001.

**Figure 3 pathogens-08-00232-f003:**
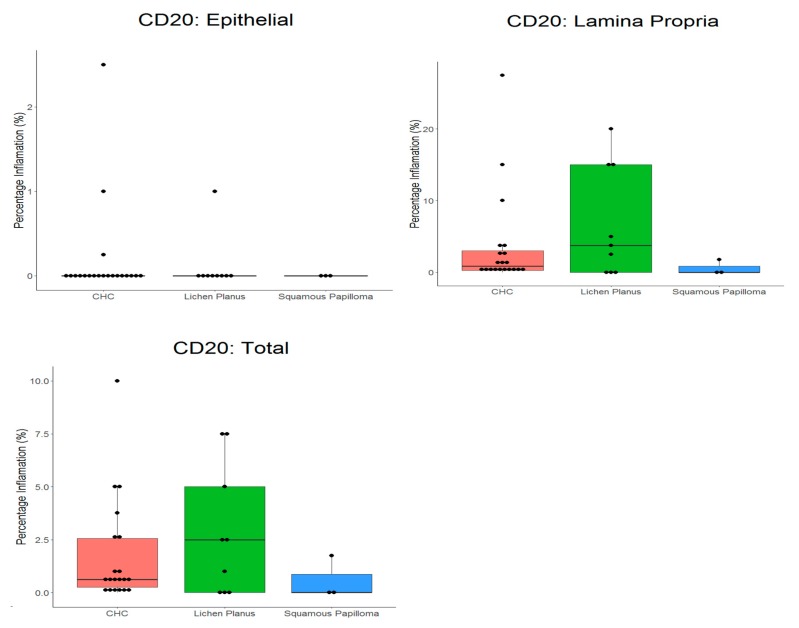
Box plot with the ‘raw’ data points superimposed, showing percentage of B cells (CD20^+^) in tissue types. CHC, chronic hyperplastic candidosis.

**Figure 4 pathogens-08-00232-f004:**
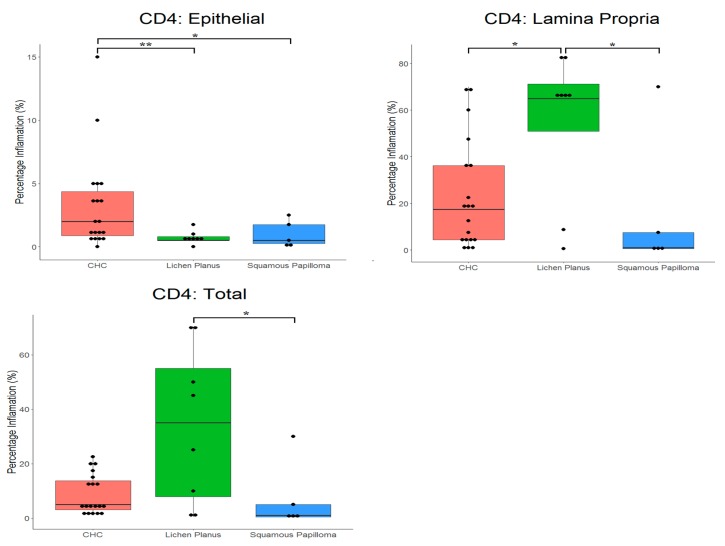
Box plot with the ‘raw’ data points superimposed, showing percentage of T helper (CD4^+^) in tissue types. CHC, chronic hyperplastic candidosis. * indicates *p* < 0.05, ** indicates *p* < 0.01.

**Figure 5 pathogens-08-00232-f005:**
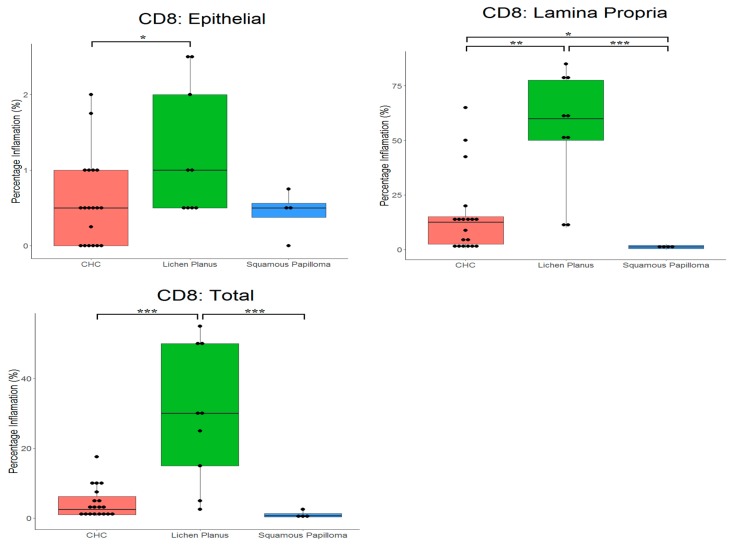
Box plots with the ‘raw’ data points superimposed, showing percentage of cytotoxic T cells (CD8^+^) in tissue types. CHC, chronic hyperplastic candidosis. * indicates *p* < 0.05, ** indicates *p* < 0.01, *** indicates *p* < 0.001.

**Figure 6 pathogens-08-00232-f006:**
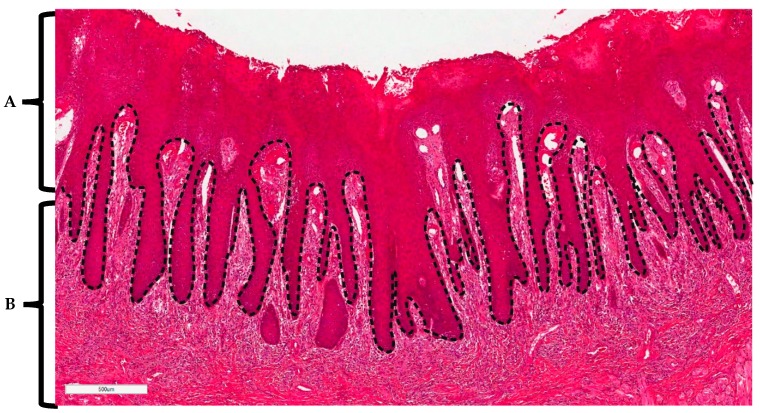
Haematoxylin and eosin stain of chronic hyperplastic candidosis illustrating delineation between (**A**) epithelium and (**B**) lamina propria/corium. Scale bar (bottom left) equals 0.5 mm.

**Table 1 pathogens-08-00232-t001:** Summary of proportions of T cells (CD3^+^) cells in tissue sections. CHC, chronic hyperplastic candidosis; LP, oral lichen planus; SqP, squamous papilloma.

		CHC (*n* = 20)	LP (*n* = 10)	SqP (*n* = 4)
Epithelium	Mean	3.9	0.9	4.9
	Median	3.8	0.8	2.0
	SD	2.6	0.6	6.9
Lamina Propria	Mean	32.6	67.2	2.9
	Median	33.8	75.0	0.6
	SD	22.2	29.2	4.7
Total	Mean	15.6	33.3	4.9
	Median	15.0	30.0	2.1
	SD	9.1	24.1	6.9

**Table 2 pathogens-08-00232-t002:** Summary of proportions of B cells (CD20^+^) in tissue sections. CHC, chronic hyperplastic candidosis; LP, oral lichen planus; SqP, squamous papilloma.

		CHC (*n* = 20)	LP (*n* = 10)	SqP (*n* = 4)
Epithelium	Mean	0.2	0.1	0.0
	Median	0.0	0.0	0.0
	SD	0.6	0.3	0.0
Lamina Propria	Mean	3.6	6.8	0.6
	Median	0.9	3.8	0.0
	SD	6.8	7.7	1.0
Total	Mean	1.8	2.9	0.6
	Median	0.6	2.5	0.0
	SD	2.5	3.1	1.0

**Table 3 pathogens-08-00232-t003:** Summary of proportions of T helper (CD4^+^) cells in tissue sections. CHC, chronic hyperplastic candidosis; LP, oral lichen planus; SqP, squamous papilloma.

		CHC (*n* = 20)	LP (*n* = 10)	SqP (*n* = 4)
Epithelium	Mean	3.2	0.7	1.0
	Median	2.0	0.5	0.5
	SD	3.8	0.5	1.1
Lamina Propria	Mean	23.1	54.6	15.9
	Median	17.5	65.0	1.0
	SD	23.3	31.8	30.4
Total	Mean	9.0	34.0	7.4
	Median	5.0	35.0	1.0
	SD	7.2	28.8	12.8

**Table 4 pathogens-08-00232-t004:** Summary of proportions of cytotoxic T cells (CD8^+^) in tissue sections. CHC, chronic hyperplastic candidosis; LP, oral lichen planus; SqP, squamous papilloma.

		CHC (*n* = 20)	LP (*n* = 10)	SqP (*n* = 4)
Epithelium	Mean	0.6	1.2	0.4
	Median	0.5	1.0	0.5
	SD	0.6	0.9	0.3
Lamina Propria	Mean	15.0	54.4	1.1
	Median	12.5	60.0	1.1
	SD	18.1	27.3	1.1
Total	Mean	4.4	29.1	1.0
	Median	2.5	30.0	0.8
	SD	4.6	19.5	1.1

**Table 5 pathogens-08-00232-t005:** Summary of oral pathology tissue types studied and their site of origin.

	Number of Each Oral Pathology Type(s) *		
Oral site of tissue	Chronic hyperplastic candidosis	Lichen planus	Squamous papiloma
Buccal mucosa	4	6	
Tongue	8	3	3
Commisure	7		
Lips	1		1
Gingiva		1	
Total	20	10	4

***** Tonsil tissue sections were included for antibody/inflammatory controls and were not used for comparison.

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
