# Peer review of "Immunohistochemical Expression Patterns of Inflammatory Cells Involved in Chronic Hyperplastic Candidosis"

_pathogens, 2019, doi:10.3390/pathogens8040232_

Round 1
Reviewer 1 Report
The introduction lacks a more detailed description of CD3+ and CD20+ and their role during a Candida infection. The reasoning why these are being studies needs to be explained in more detail. The introduction would also benefit from more detail on the role of antibodies during Candida infections.
The results section for "immunohistochemical analysis" needs a better explanation of what the purpose of figure 1 is and why it is useful for the paper. Additionally, more detail on what is present is required.
The paper would also benefit from a brief description of where the tissues came from during the result section since this is only discussed in methods.
The different tissue samples need to be better explained during the results section (i.e. which ones are controls for what). This is only in the results section and would serve the reader to find that information earlier.
Figures 1, 3, 4, and 5 would be better depicted as bar graphs. Additionally, the control group for each of these should also a higher n.
Author Response
Re: Manuscript ID: pathogens-621766; Immunohistochemical expression patterns of inflammatory cells involved in chronic hyperplastic candidosis; Williams et al
Many thanks for the time and effort that the reviewer has afforded to improve the manuscript submitted for consideration for publication in the journal Pathogens. We are very grateful for the comments provided and have amended the text (highlighted in yellow in the revised document) accordingly, as well as detailed our responses below.
Reviewer 1.
The introduction lacks a more detailed description of CD3+ and CD20+ and their role during a Candida infection. The reasoning why these are being studies needs to be explained in more detail. The introduction would also benefit from more detail on the role of antibodies during Candida infections.
Response: The text has been modified to include a more detailed description of CD3+ and CD20+ cells and their role in Candida infection. We have also added some more information regarding antibodies in Candida infection. Please see lines 61-82.
The results section for "immunohistochemical analysis" needs a better explanation of what the purpose of figure 1 is and why it is useful for the paper. Additionally, more detail on what is present is required.
Response: The primary purpose of figure 1 is to illustrate typical immunohistochemical staining for each antibody and tissue type. It shows to the reader that the antibody staining functioned well and gives an indication where the stained cells were in the different tissues.
The paper would also benefit from a brief description of where the tissues came from during the result section since this is only discussed in methods.
Response: We have now added Table 5 that provides this information.
The different tissue samples need to be better explained during the results section (i.e. which ones are controls for what). This is only in the results section and would serve the reader to find that information earlier.
Response: Lines 88-91 now restate the tissue types used and their purpose.
Figures 1, 3, 4, and 5 would be better depicted as bar graphs. Additionally, the control group for each of these should also a higher n.
Thank you for this comment. The control group here was just an antibody control and not used for comparison, we have therefore removed this data from the figures. We discussed the presentation of these figures at length with our statistician, Dr Farnell. Two recent articles (Weissgerber et al PLoS Biol 13(4): e1002128 (2015); Weissgerber et al. J. Biol. Chem. (2017) 292(50) 20592–20598 (2017)) have highlighted the deficiencies of using bar graphs to represent summary statistics of data, especially when small sample sizes in any group are extremely small. We have used recommendations from these articles, namely, that a box plot with the "raw" data points superimposed is a good way of visualising the data when we cannot guarantee that it is normally distributed. Such figures are increasingly common in biology journals nowadays, especially when sample sizes are small. We hope that these figures are now acceptable to the referee.
Reviewer 2 Report
This is an interesting article on an important topic. The authors show the infiltration and abundance of T cells (CD4 and CD8) as well as B cells during chronic hyperplastic candidosis (CHC). Using immunohistochemistry it was concluded that during CHC immunity is primarily driven by the T helper cells. I have some minor points which should be addressed
-Figure 1 needs scale bars, also the authors should indicate how the authors distinguished between epithelium, corium, etc. Furthermore what means control in this case? No infection and no inflammation? This has to be better presented.
-l. 51 ff: epithelium senses fungal outgrowth via receptor activation (PMID: 29133884) and induces antifungal mechanisms (PMID: 20833374, 29371576)
Author Response
Re: Manuscript ID: pathogens-621766; Immunohistochemical expression patterns of inflammatory cells involved in chronic hyperplastic candidosis; Williams et al
Many thanks for the time and effort that the reviewer has afforded to improve the manuscript submitted for consideration for publication in the journal Pathogens. We are very grateful for the comments provided and have amended the text (highlighted in yellow in the revised document) accordingly, as well as detailed our responses below.
Reviewer 2.
This is an interesting article on an important topic. The authors show the infiltration and abundance of T cells (CD4 and CD8) as well as B cells during chronic hyperplastic candidosis (CHC). Using immunohistochemistry it was concluded that during CHC immunity is primarily driven by the T helper cells. I have some minor points which should be addressed.
-Figure 1 needs scale bars, also the authors should indicate how the authors distinguished between epithelium, corium, etc. Furthermore what means control in this case? No infection and no inflammation? This has to be better presented.
Response: Following discussion with our Oral Pathologist (Dr A Jones) we feel that given the magnification of the images in figure 1, scale bars will be very difficult to illustrate. However, we have now included a larger image (figure 6) which includes a scale bar for reference too. If possible we would like to retain figure 1. The primary purpose of figure 1 is to illustrate typical immunohistochemical staining for each antibody and tissue type. It shows to the reader that the antibody staining functioned well and gives an indication where the stained cells were in the different tissues..
The ‘control’ was a tonsil tissue and included to confirm antibody staining of inflammatory cells. We have however now removed from our comparative figures. The value of the other tissues in this comparison have now been highlighted in the text.
-l. 51 ff: epithelium senses fungal outgrowth via receptor activation (PMID: 29133884) and induces antifungal mechanisms (PMID: 20833374, 29371576)
Response: It is certainly possible that epithelial cells respond to the invading Candida by generating cytokines which could serve to recruit immune functioning cells. We have added some text to the introduction (lines 61-64) to highlight this along with supporting references.
Round 2
Reviewer 1 Report
Thank you for the update. This manuscript will be of great interest to the fungal community.